# Unsupervised Speech Separation Using Mixtures of Mixtures

**Scott Wisdom** [1]   **Efthymios Tzinis** [* 1 2]   **Hakan Erdogan** [1]   **Ron J. Weiss** [1]   **Kevin Wilson** [1]   **John R. Hershey** [1]

## Abstract

Supervised approaches to single-channel speech separation rely on synthetic mixtures, so that the individual sources can be used as targets. Good performance depends upon how well the synthetic mixture data match real mixtures. However, matching synthetic data to the acoustic properties and distribution of sounds in a target domain can be challenging. Instead, we propose an unsupervised method that requires only single-channel acoustic mixtures, without ground-truth source signals. In this method, existing mixtures are mixed together to form a *mixture of mixtures*, which the model separates into latent sources. We propose a novel loss that allows the latent sources to be remixed to approximate the original mixtures. Experiments show that this method can achieve competitive performance on speech separation compared to supervised methods. In a semi-supervised learning setting, our method enables domain adaptation by incorporating unsupervised mixtures from a matched domain. In particular, we demonstrate that significant improvement to reverberant speech separation performance can be achieved by incorporating reverberant mixtures.

## 1. Introduction

Audio perception is fraught with a fundamental problem: individual sounds are convolved with unknown acoustic reverberation functions and mixed together at the acoustic sensor in a way that is impossible to disentangle without prior knowledge of the source characteristics. It is a hallmark of human hearing that we are able to hear the nuances of different sources, even when presented with a monaural mixture of sounds. Significant progress has been made on extracting estimates of each source from single-channel recordings,

using supervised deep learning methods. These techniques have been applied to tasks such as speaker-independent enhancement (separation of speech from nonspeech interference) (Huang et al., 2014; Weninger et al., 2015) and speech separation (separation of speech from speech) (Hershey et al., 2016; Isik et al., 2016; Yu et al., 2017).

These approaches have used supervised training, in which ground-truth source waveforms are considered targets for various loss functions. Deep clustering (Hershey et al., 2016) is an embedding-based approach that implicitly represents the assignment of elements of a mixture, such as time-frequency bins of a spectrogram, to sources in a way that is independent of any ordering of the sources. In permutation-invariant training (Isik et al., 2016; Yu et al., 2017), the model explicitly outputs the signals in an arbitrary order, and the loss function finds the permutation of that order that best matches the estimated signals to the references, i.e. treating the problem as a set prediction task. In both cases the ground-truth signals are inherently part of the loss.

A major problem with supervised training for source separation is that it is not feasible to record both the mixture signal and the individual ground-truth source signals in a real acoustic environment, because source recordings are contaminated by cross-talk. Therefore supervised training has relied on synthetic mixtures created by adding up isolated ground-truth sources, with or without a simulation of the acoustic environment. Although supervised training has been effective in training models that perform well on data that match the same distribution of mixtures, they fare poorly when there is mismatch in the distribution of sound types (Manilow et al., 2019), or in acoustic conditions such as reverberation (Maciejewski et al., 2018). It is difficult to match the characteristics of a real dataset because the distribution of source types and room characteristics may be unknown and difficult to estimate, data of every source type in isolation may not be readily available, and accurately simulating realistic acoustics is challenging.

One approach to avoiding these difficulties is to use acoustic mixtures from the target domain, without references, directly in training. To that end, weakly supervised training has been proposed to substitute the strong labels of source references with another modality such as class labels, visual features, or spatial information. In (Pishdadian et al., 2019)

---

*Work done during an internship at Google. [1]Google Research [2]Department of Computer Science, University of Illinois at Urbana-Champaign, IL, USA. Correspondence to: Scott Wisdom <scottwisdom@google.com>.

*Published at the workshop on Self-supervision in Audio and Speech at the 37th International Conference on Machine Learning*, Vienna, Austria. Copyright 2020 by the author(s).

class labels were used as a substitute for signal-level losses. The spatial locations of individual sources, which can be inferred from multichannel audio, has also been used to guide learning of single-channel separation (Tzinis et al., 2019; Seetharaman et al., 2019; Drude et al., 2019). Visual input corresponding to each source has been used to supervise the extraction of the corresponding sources in (Gao & Grauman, 2019), where the targets included mixtures of sources, and the mapping between source estimates and mixture references was given by the video correspondence. Because these approaches rely on multimodal training data containing extra input modalities, they cannot be used in settings where only single-channel audio is available.

We propose a novel unsupervised training framework that requires only single-channel acoustic mixtures. This framework is related to permutation-invariant training (PIT) (Yu et al., 2017), in which the permutation used to match source estimates to source references is relaxed to allow summation over some of the sources. In our proposed *mixture invariant training* (MixIT), instead of single-source references, we use mixtures from the target domain as references, forming the input to the separation model by summing together these mixtures to form a mixture of mixtures. The model is trained to separate this input into a variable number of latent source estimates, such that the separated sources can be remixed to approximate the original mixtures.

**Contributions**: (1) we propose the first purely unsupervised learning method that is effective for audio-only single-channel speech separation and find that it can achieve competitive performance with supervised methods; (2) we provide extensive experiments with cross-domain adaptation to show the effectiveness of MixIT for adaptation to different reverberation characteristics in semi-supervised settings.

## 2. Relation to previous work

Discriminative source separation models generate synthetic mixtures from isolated sources which are also used as training targets. Early methods posed the problem in terms of time-frequency mask estimation, and considered restrictive cases such as speaker-dependent models, and class-specific separation, e.g. speech versus music (Huang et al., 2014), or noise (Weninger et al., 2015). However, more general speaker-independent speech separation, and class-independent universal sound separation (Kavalerov et al., 2019; Tzinis et al., 2020) are now addressed using methods such as deep clustering (Hershey et al., 2016) and PIT (Yu et al., 2017). These frameworks handle the output permutation problem caused by the lack of a unique source class for each output. Recent state-of-the-art models have shifted from mask-based recurrent networks to time-domain convolutional networks (Luo & Mesgarani, 2019). MixIT follows this trend and uses a signal-level discriminative loss. The

framework can be used with any architecture; in this paper we use a modern time-convolutional network. Unlike supervised approaches, MixIT can use raw-mixtures as references and enable training directly on target-domain mixtures for which ground-truth source signals cannot be obtained. Previous methods proposed domain adaptation schemes by using adversarial training to learn domain-invariant intermediate network activations (Ganin et al., 2016; Tzeng et al., 2017) or train student and teacher models to predict consistent separated estimates from supervised and unsupervised mixtures (Lam et al., 2020). In contrast, MixIT not only works under purely unsupervised settings, but it also enables a seamless semi-supervised scheme to train a single network with both supervised and unsupervised losses.

Similar to MixIT, (Gao & Grauman, 2019) uses *mixtures of mixtures* (MoMs) as input, and sums over estimated sources to match the target mixtures, using the *co-separation loss*. However, this loss does not identify correspondence between sources and mixtures, since that is established by the supervising video inputs, each of which is assumed to correspond to one source. In MixIT this is handled in an unsupervised manner, by finding the best correspondence between sums of sources and the reference mixtures without using other modalities, making the proposed methods the first fully unsupervised separation work using MoMs.

Also related is *adversarial unmix-and-remix* (Hoshen, 2019), which separates linear image mixtures in a GAN framework, with the discriminator operating on mixtures rather than single sources. Mixtures are separated, and the resulting sources are remixed to form new mixtures which are pushed to match the distribution of the original inputs. The authors reported good separation results on image mixtures, but their method failed on audio mixtures. In contrast, MixIT avoids the difficulty of saddle-point optimization associated with GANs and works well on audio mixtures. MixIT uses MoMs and relies on generalization to work on single mixtures while unmix-and-remix has the advantage of being trained with the original mixtures. However, unmix-and-remix could be combined with MixIT in future work.

## 3. Method

We generalize the permutation-invariant training framework to operate directly on unsupervised mixtures, as illustrated in Figure 1. Formally, a supervised separation dataset is comprised of pairs of input mixtures $\mathbf{x} = \sum_{n=1}^{N} \mathbf{s}_n$ and their constituent sources $\mathbf{s}_n \in \mathbb{R}^T$, where each mixture contains up to $N$ sources with $T$ time samples each. Without loss of generality, for the mixtures that contain only $N' < N$ sources we assume that $\mathbf{s}_n = \mathbf{0}$ for $N' < n \leq N$. An unsupervised dataset only contains input mixtures without underlying reference sources. We assume that the maximum number of sources present in any mixture is known.

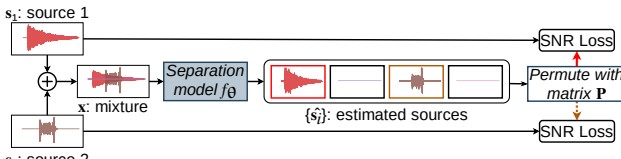

(a) Supervised permutation invariant training (PIT).

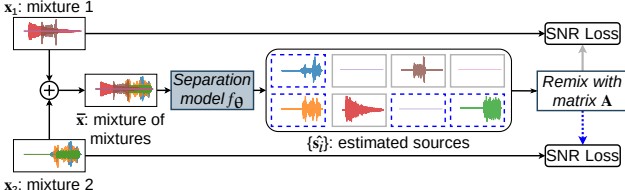

(b) Unsupervised mixture invariant training (MixIT).

Figure 1: Overview of (a) PIT separating a two-source mixture into up to four sources and (b) MixIT separating a mixture of mixtures into up to eight sources. Arrow color indicates best match between estimates and references.

## 3.1. Permutation invariant training

In the supervised case we are given a mixture $\mathbf{x}$ and its corresponding sources $\mathbf{s}$ to train on. The input mixture $\mathbf{x}$ is fed through a separation model $f_{\boldsymbol{\theta}}$ with parameters $\boldsymbol{\theta}$. The model estimates $M$ sources: $\hat{\mathbf{s}} = f_{\boldsymbol{\theta}}(\mathbf{x}) \in \mathbb{R}^{M \times T}$, where $M = N$ is the maximum number of sources co-existing in any given mixture drawn from the supervised dataset. The supervised separation loss can be written as:

$$\mathcal{L}_{\mathrm{PIT}}(\mathbf{s}, \hat{\mathbf{s}}) = \min_{\mathbf{P}} \sum_{m=1}^{M} \mathcal{L}(\mathbf{s}_m, [\mathbf{P}\hat{\mathbf{s}}]_m), \qquad (1)$$

where $\mathbf{P}$ is an $M \times M$ permutation matrix and $\mathcal{L}$ is a signal-level loss function. There is no predefined ordering of the source signals. Instead, the loss is computed using the permutation which gives the best match between ground-truth reference sources $\mathbf{s}$ and estimated sources $\hat{\mathbf{s}}$.

The loss function between a reference $\mathbf{y} \in \mathbb{R}^T$ and estimate $\hat{\mathbf{y}} \in \mathbb{R}^T$ is the negative thresholded signal-to-noise ratio:

$$\mathcal{L}(\mathbf{y}, \hat{\mathbf{y}}) = -10 \log_{10} \frac{\|\mathbf{y}\|^2}{\|\mathbf{y} - \hat{\mathbf{y}}\|^2 + \tau \|\mathbf{y}\|^2}, \qquad (2)$$

where $\tau = 10^{-\mathrm{SNR}_{\max}/10}$ acts as a soft threshold that clamps the loss at $\mathrm{SNR}_{\max}$. This threshold prevents examples that are already well-separated from dominating the gradients within a training batch. We use $\mathrm{SNR}_{\max} = 30$.

## 3.2. Mixture invariant training

PIT requires knowledge of the ground truth source signals $\mathbf{s}$, and therefore cannot directly leverage unsupervised data where only mixtures $\mathbf{x}$ are observed. MixIT overcomes this

problem as follows. Consider two mixtures $\mathbf{x}_1$ and $\mathbf{x}_2$ are drawn without replacement from an unsupervised dataset where each one is comprised of up to $N$ underlying sources (any number of mixtures could be used, but here we use two for simplicity). The mixture of mixtures is formed by adding them together: $\overline{\mathbf{x}} = \mathbf{x}_1 + \mathbf{x}_2$. The separation model $f_{\boldsymbol{\theta}}$ takes $\overline{\mathbf{x}}$ as input, and estimates $M = 2N$ latent source signals. In this way we make sure that the model is always capable of estimating enough sources for any $\overline{\mathbf{x}}$. The unsupervised MixIT loss is computed between the estimated sources $\hat{\mathbf{s}}$ and the input mixtures $\mathbf{x}_1, \mathbf{x}_2$ as follows:

$$\mathcal{L}_{\mathrm{MixIT}}(\mathbf{x}_1, \mathbf{x}_2, \hat{\mathbf{s}}) = \min_{\mathbf{A}} \sum_{i=1}^{2} \mathcal{L}(\mathbf{x}_i, [\mathbf{A}\hat{\mathbf{s}}]_i), \qquad (3)$$

where $\mathcal{L}$ is the same signal-level loss used in PIT (2) and the *mixing matrix* $\mathbf{A} \in \mathbb{B}^{2 \times M}$ is constrained to the set of $2 \times M$ binary matrices where each column sums to 1. Thus, each latent source $\hat{\mathbf{s}}_m$ can only be used once, and is assigned to either $\mathbf{x}_1$ or $\mathbf{x}_2$. MixIT minimizes the total loss between mixtures $\mathbf{x}$ and remixed latent sources $\mathbf{A}\hat{\mathbf{s}}$ by choosing the best match between sources and mixtures (analogous to PIT). In practice, we optimize over $\mathbf{A}$ using an exhaustive $\mathcal{O}(2^M)$ search, although more efficient methods are possible.

There is an implicit assumption in MixIT that the sources are additive, and that they are independent of each other in the original mixtures $\mathbf{x}_1$ and $\mathbf{x}_2$, in the sense that there is no information in $\overline{\mathbf{x}}$ about which sources belong to which mixtures. The two mixtures $\mathbf{x} \in \mathbb{R}^{2 \times T}$ are assumed to result from mixing unknown sources $\mathbf{s}^* \in \mathbb{R}^{P \times T}$ using an unknown $2 \times P$ mixing matrix $\mathbf{A}^*$: $\mathbf{x} = \mathbf{A}^* \mathbf{s}^*$. If the network could infer which sources belong together in the references, and hence knew the mixing matrix $\mathbf{A}^*$ (up to a left permutation), then the $M$ source estimates, $\hat{\mathbf{s}} \in \mathbb{R}^{M \times T}$ could minimize the loss (3) without separating all the sources (i.e., by under-separating). That is, for a known mixing matrix $\mathbf{A}^*$, the loss (3) could be minimized, for example, by the estimate $\hat{\mathbf{s}} = \mathbf{C}^+ \mathbf{A}^* \mathbf{s}^*$, with $\mathbf{C}^+$ the pseudoinverse of a $2 \times M$ mixing matrix $\mathbf{C}$ such that $\mathbf{C}\mathbf{C}^+ = \mathbf{I}$, at $\mathbf{A} = \mathbf{C}$, since $\mathbf{C}\hat{\mathbf{s}} = \mathbf{C}\mathbf{C}^+ \mathbf{A}^* \mathbf{s}^* = \mathbf{x}$. However, if the sources are independent, then the network cannot infer the mixing matrix that produced the reference mixtures. Nevertheless, the loss can be minimized with a single set of estimates, regardless of the mixing matrix $\mathbf{A}^*$, by separating all of the sources. That is, the estimated sources must be within a mixing matrix $\mathbf{B} \in \mathbb{B}^{P \times M}$ of the original sources, $\mathbf{s}^* = \mathbf{B}\hat{\mathbf{s}}$, so that (3) is minimized at $\mathbf{A} = \mathbf{A}^* \mathbf{B}$, for any $\mathbf{A}^*$. Hence, the lack of knowledge about which sources belong to which mixtures encourages the network to separate the sources as much as possible. Note that when $M > P$, the network can produce more estimates than there are sources (i.e., over-separate). In this work, semi-supervised training may help with this, and future work will address methods to penalize over-separation in the fully unsupervised case.

## 3.3. Semi-supervised training

When trained with $M$ isolated reference sources, i.e. with full supervision, the MixIT loss is equivalent to PIT. Specifically, input mixtures $\mathbf{x}_i$ are replaced with ground-truth reference sources $\mathbf{s}_m$ and the mixing matrix $\mathbf{A}$ becomes an $M \times M$ permutation matrix $\mathbf{P}$. This makes it straightforward to combine both losses to perform semi-supervised learning. In essence, each training batch contains $p\%$ unsupervised mixtures, for which we do not know the constituent sources and use the MixIT loss (3), and the remainder supervised examples, for which we use the PIT loss (1).

# 4. Experiments

Our separation model $f_{\boldsymbol{\theta}}$ consists of a learnable convolutional basis transform that produces mixture basis coefficients. These are processed by an improved time-domain convolutional network (TDCN++) (Kavalerov et al., 2019), similar to ConvTasNet (Luo & Mesgarani, 2019). This network predicts $M$ masks with values between 0 and 1 and the same size as the basis coefficients. The $M$ separated waveforms are produced by overlapping and adding the masks elementwise multiplied with the coefficients. A mixture consistency projection (Wisdom et al., 2019) is applied to constrain separated sources to add up to the input mixture. See Appendix A for architecture and training details.

Separation performance is measured using scale-invariant signal-to-noise ratio (SI-SNR) (Le Roux et al., 2019). SI-SNR measures fidelity between a signal $\mathbf{y}$ and its estimate $\hat{\mathbf{y}}$ within an arbitrary scale factor:

$$\text{SI-SNR}(\mathbf{y}, \hat{\mathbf{y}}) = 10 \log_{10} \frac{\|\alpha \mathbf{y}\|^2}{\|\alpha \mathbf{y} - \hat{\mathbf{y}}\|^2}, \qquad (4)$$

where $\alpha = \text{argmin}_a \|a\mathbf{y} - \hat{\mathbf{y}}\|^2 = \mathbf{y}^T \hat{\mathbf{y}} / \|\mathbf{y}\|^2$. Generally we report SI-SNR improvement (SI-SNRi), which is the difference between the SI-SNR of each source estimate after processing, and the SI-SNR obtained using the input mixture as the estimate for each source. In our evaluations, mixtures can contain fewer than the $M$ sources output by the model. To handle this, we zero-pad the references to $M$ sources, permute these references to match the separated sources, and average SI-SNRi over non-zero references.

For speech separation experiments, we use the WSJ0-2mix (Hershey et al., 2016), sampled at 8 kHz or 16 kHz, and Libri2Mix (Cosentino et al., 2020) datasets, sampled at 16 kHz. We also employ the reverberant spatialized versions of WSJ0-2mix (Wang et al., 2018) and a reverberant version of Libri2Mix we created. Both datasets consist of utterances from male and female speakers drawn from either the Wall Street Journal (WSJ0) corpus or from LibriSpeech (Panayotov et al., 2015). Reverberant versions are created by convolving utterances with room impulse responses gen-

erated by a room simulator employing the image method (Allen & Berkley, 1979). WSJ0-2mix and the train-360-clean split of Libri2Mix provide 30 hours and 364 hours of training mixtures, respectively. Note that for WSJ0-2mix individual source utterances are drawn with replacement.

We sweep the amount of supervised versus unsupervised data for both the anechoic and reverberant versions of WSJ0-2mix. The proportion $p$ of unsupervised data from the same domain is swept from 0% to 100% where supervised training uses the PIT separation loss (2) between ground-truth references and separated sources, and unsupervised training only uses the mixtures using MixIT (3) with the same separation loss (2) between mixtures and remixed separated sources. In both cases, the input to the separation model is a mixture of two mixtures. For training, 3 second clips are used for WSJ0-2mix, and 10 second clips for Libri2Mix.

We try two variants of this task: mixtures that always contain two speakers (2-source) such that MoMs always contain four sources, and mixtures containing either one or two speakers (1-or-2-source) such that MoMs contain two to four sources. Note that the network always has four outputs. Evaluation always uses single mixtures of two sources. To determine if unsupervised data can help with domain mismatch, we also consider using supervised data from a mismatched domain, by incorporating supervised anechoic data (from the same task) when using the MixIT loss on reverberant mixtures and vice versa. This simulates the realistic training scenario for sound separation systems, where real acoustic mixtures from a target domain are available without reference waveforms and synthetic supervised data must be created to match the distribution of the real data. It is difficult to perfectly match the real data distribution, so synthetic supervised data will inevitably have some mismatch to the target domain.

The results on anechoic and reverberant WSJ0-2mix and Libri2Mix are shown in Figure 2. First, notice that reverberant data is more challenging to separate because reverberation smears out the spectral energy of sources over time, and thus all models achieve lower performance on reverberant data. Two-source mixture trained models tends to do less well compared to the 1-or-2-source variants. One difference with the 1-or-2-source setup is that the model observes some inputs that have two sources, which matches the evaluation. Another difference is that as references, the 1-source mixtures act as supervised examples.

Notice that for both anechoic and reverberant data, completely unsupervised training with MixIT (rightmost points)[1] achieves performance on par with supervised training (leftmost points) with 1-or-2-source mixtures. For 2-source mixtures, unsupervised performance is worse by up to 3 dB

---

[1]The rightmost matched and mismatched points use identical training data since no supervision is used. Small performance differences reflect randomness in model initialization and training.

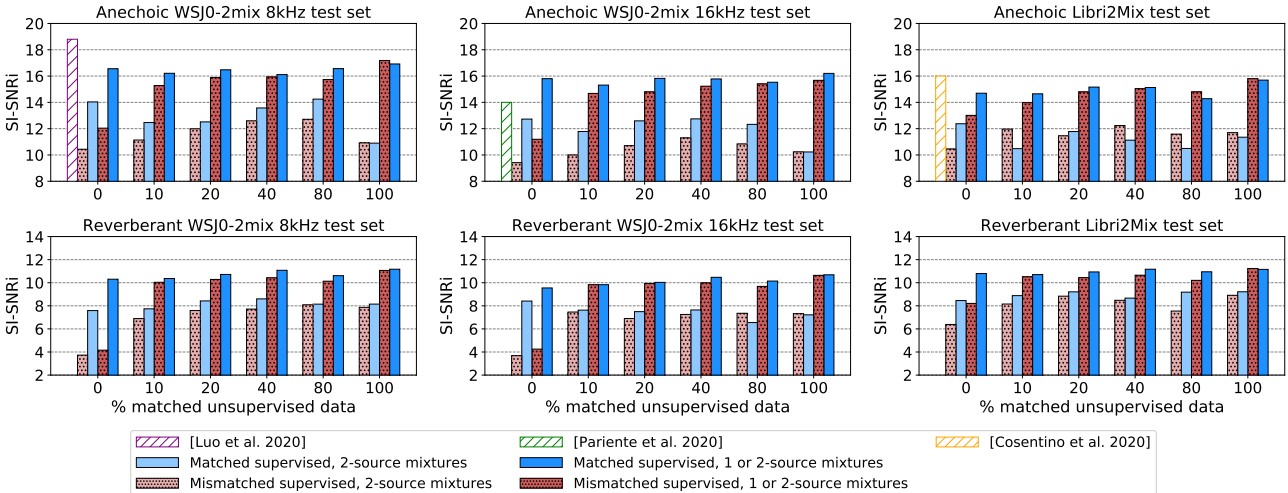

Figure 2: Sweeping proportion of matched unsupervised training examples with matched or mismatched supervised examples on WSJ0-2mix 8kHz (left), WSJ0-2mix 16kHz (middle), and Libri2Mix (right). The leftmost points in each plot correspond to 100% supervision using PIT, and the rightmost points are fully unsupervised using MixIT.

compared to fully or semi-supervised on anechoic data, while performance is more comparable on reverberant data. However, even a small amount of supervision (80% unsupervised) dramatically improves anechoic SI-SNRi. When the supervised data is mismatched, adding a small amount of unsupervised data (10%) from a matched domain drastically improves performance: using mismatched anechoic supervised data and matched reverberant unsupervised data, we observe boosts of 2-3 dB for 2-source mixtures on all datasets. For 1-to-2-source mixtures, performance increases by about 6 dB on WSJ0-2mix and 2.5 dB for Libri2Mix.

Although our primary focus is on less supervised learning, MixIT models are competitive on anechoic datasets with state-of-the-art approaches that do not exploit speaker identity information. Fig. 2 includes the best reported numbers for 8 and 16 kHz WSJ0-2mix (Luo et al., 2020; Pariente et al., 2020), and Libri2Mix (Cosentino et al., 2020).

### 4.1. Discussion

The experiments show the effectiveness of MixIT and that unsupervised domain adaptation always helps: matched fully unsupervised training is always better than mismatched fully supervised training, often by a large margin. To the best of our knowledge, this is the first single-channel purely unsupervised separation method which obtains comparable performance to state-of-the-art supervised approaches.

In some of the experiments reported here, the data preparation has some limitations. The WSJ0-2mix data have the property that each unique source may be repeated across multiple mixture examples, whereas Libri2Mix uses unique sources in every mixture. Such re-use of source signals is not a problem for ordinary supervised separation, but in the

context of MixIT, there is a possibility that the model may abuse this redundancy. In particular in the 1-or-2 source case, this raises the chance that each source appears as a reference, which could make the unsupervised training act more like supervised training. However, the unsupervised performance on Libri2Mix, which does not contain redundant sources, parallels the WSJ0-2mix results and shows that if there is a redundancy loophole to be exploited in some cases, it is not needed for good performance.

An ultimate goal is to evaluate separation on real mixture data; however, this remains challenging because of the lack of ground truth. As a proxy, future experiments may use recognition or human listening as a measure of separation.

## 5. Conclusion

We have presented MixIT, a new paradigm for training separation models in a completely unsupervised manner where ground-truth source references are not required. On a speech separation task, we demonstrated that MixIT can approach the performance of supervised PIT, and is especially helpful in a semi-supervised setup to adapt to mismatched domains. More broadly, MixIT opens new lines of research where large amounts of previously untapped in-the-wild data can be leveraged to train sound separation systems.

## Acknowledgements

The idea of using mixtures of mixtures came about during the 2015 JHU CLSP Jelinek Summer Workshop at University of Washington. Special thanks goes to Jonathan Le Roux for prior discussions of other methods using MoMs, and to Aren Jansen for helpful comments on the manuscript.

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

## A. Separation model architecture

In Table 1, we describe the separation network architecture using a TDCN++ (Kavalerov et al., 2019). As compared to the original ConvTasNet method (Luo & Mesgarani, 2019), the changes to the model include the following:

- Instead of global layer norm, which averages statistics over frames and channels, the TDCN++ uses instance norm, also known as feature-wise global layer norm (Kavalerov et al., 2019). This mean-and-variance normalization is performed separately for each convolution channel across frames, with trainable scalar bias and scale parameters.

- Additional skip-residual connections from the outputs of earlier residual blocks to the inputs of the later residual blocks. A skip-residual connection includes a transformation in the form of a dense layer with bias of the block outputs and all paths from residual connections are summed with the regular block input coming from the previous block. Note that all dense layers in the model include bias terms.

- A scalar scale parameter is multiplied after each dense layer stage, which is an over-parametrization trick that improves convergence. The scale parameters for the second dense layer in layer $i$ are initialized using exponential decay in the form of $0.9^i$. All other scales are initialized to 1.0. This initial scaling controls the contribution of each block into the residual sum. It also causes the initial blocks train faster and the later blocks to train slower, which is reminiscent of layer-wise training.

As mentioned in the text, we also apply a mixture consistency projection (Wisdom et al., 2019) to the resulting separated waveforms, which projects them such that they sum up to the original mixture. This projection solves the following optimization problem to find mixture consistency separated sources $\hat{\mathbf{s}}$ given initial separated sources $\underline{\mathbf{s}}$ separated by the model from a mixture $\mathbf{x}$:

$$\begin{aligned} \underset{\hat{\mathbf{s}} \in \mathbb{R}^{M \times T}}{\text{minimize}} \quad & \frac{1}{2} \sum_m \|\hat{\mathbf{s}}_m - \underline{\mathbf{s}}_m\|^2 \\ \text{subject to} \quad & \sum_m \hat{\mathbf{s}}_m = \mathbf{x}. \end{aligned} \tag{5}$$

The projection operation is the closed-form solution of this problem:

$$\hat{\mathbf{s}}_m = \underline{\mathbf{s}}_m + \frac{1}{M}(\mathbf{x} - \sum_{m'} \underline{\mathbf{s}}_{m'}), \tag{6}$$

which is differentiable and can simply be applied as a final layer to the initial separated sources $\underline{\mathbf{s}}$.

## B. Training details

For each task, we train all models to 200k steps, evaluating a checkpoint every 10 minutes. For evaluation on the test set, we select the checkpoint with the highest validation score. As mentioned in the text, all models are trained with batch size 256 with the Adam optimizer (Kingma & Ba, 2015) using a learning rate of $10^{-3}$ on 4 Google Cloud TPUs (16 chips).

## C. Ablations

In order to evaluate the contribution of different components of the proposed model we compare several variations trained on WSJ0-2mix with two-source mixtures: disabling mixture consistency, and varying $\text{SNR}_{\max}$. Performance is reported on the validation set after 200k training steps.

**Mixture consistency** We observed modest improvement of 0.5 dB SI-SNRi by incorporating mixture consistency (6) versus not.

**SNR threshold** Performance is not very sensitive to $\text{SNR}_{\max}$ as long as it is 20 dB or larger, as shown in Table 2.

**Zero source loss** For speech separation tasks using 1-to-2-source mixtures, the separation model needs to be able to output near-zero signals for "inactive" source slots. For separated signals that align to all-zeros reference source, we tried using a variation on the negative SNR loss function (2), where the mixture signal $\mathbf{x}$ instead of the source signal $\mathbf{s}$ is used to determine the soft-thresholding, where we still set $\tau$ corresponding to $\text{SNR}_{\max}$ of 30 dB:

$$\mathcal{L}_0(\mathbf{s} = \mathbf{0}, \hat{\mathbf{s}}, \mathbf{x}) = 10 \log_{10} \left( \|\hat{\mathbf{s}}\|^2 + \tau\|\mathbf{x}\|^2 \right), \tag{7}$$

which means the loss will be clipped when the power of the separated signal drops 30 dB below the power of the mixture signal.

For WSJ0-2mix, where the models are trained on mixtures of 1-to-2-source mixtures, and evaluated on single mixtures from the validation set. Using the additional zero source loss results in a SI-SNRi of 14.3 dB, while not using it leads to a SI-SNRi of 15.9 dB. Thus, incorporating $\mathcal{L}_0$ decreases SI-SNRi, and we did not use this loss to train our models.

## D. Audio examples

Audio demos for speech separation on anechoic and reverberant WSJ0-2mix 16 kHz and Libri2Mix are provided at https://universal-sound-separation. github.io/unsupervised_speech_ separation/.

Table 1: Separation network with TDCN++ architecture configuration. Variables are number of encoder basis coefficients $N = 256$, encoder basis kernel size $L$, which is 40 for 16 kHz data and 20 for 8 kHz data, number of waveform samples $T$, number of coefficient frames $F$, and number of separated sources $M$.

| Module name | Operation | Output shape | Kernel size | Dilation | Stride |
|---|---|---|---|---|---|
| Waveform | Input | $T \times 1$ | – | – | – |
| Encoder | Conv | $F \times N$ | $1 \times L \times N$ | 1 | $L/2$ |
| Coeffs | Intermediate | $F \times N$ | – | – | – |
| Initial bottleneck | ReLU | $F \times N$ | – | – | – |
| | Dense | $F \times 256$ | $N \times 256$ | 1 | 1 |
| $i$-th separable dilated conv block (x32) | Input | $F \times 256$ | Previous block output + sum of skip-residual inputs | | |
| | Dense | $F \times 512$ | $256 \times 512$ | – | – |
| with skip-residual | Scale | $F \times 512$ | $1 \times 1$ | – | – |
| connections b/w blocks: | PReLU | $F \times 512$ | – | – | – |
| $i \rightarrow i+1$, | Instance norm | $F \times 512$ | – | – | – |
| $0 \rightarrow 8, 0 \rightarrow 16, 0 \rightarrow 24,$ | Depthwise conv | $F \times 512$ | $512 \times 3 \times 1$ | $2^{\mathrm{mod}(i,8)}$ | 1 |
| $8 \rightarrow 16, 8 \rightarrow 24,$ | PReLU | $F \times 512$ | – | – | – |
| $16 \rightarrow 24,$ | Instance norm | $F \times 512$ | – | – | – |
| | Dense | $F \times 256$ | $512 \times 256$ | – | – |
| | Scale | $F \times 512$ | $1 \times 1$ | – | – |
| Final bottleneck | Dense | $F \times 256$ | $512 \times 256$ | – | – |
| Perform masking | Dense | $F \times M \cdot N$ | $256 \times M \cdot N$ | – | – |
| | Sigmoid | $F \times M \cdot N$ | – | – | – |
| | Reshape | $F \times M \times N$ | – | – | – |
| | Multiply | $F \times M \times N$ | Multiply with $F \times 1 \times N$ coeffs | | |
| Decoder | Transposed conv | $T \times M$ | $L \times N \times 1$ | 1 | $L/2$ |
| Separated waveforms | Output | $T \times M$ | – | – | – |

Table 2: SI-SNRi in dB as a function of $\mathrm{SNR_{max}}$ for unsupervised MixIT on WSJ0-2mix 2-source mixtures.

| $\mathrm{SNR_{max}}$ | 10 | 20 | 30 | 40 | 50 |
|---|---|---|---|---|---|
| SI-SNRi | 13.1 | 13.8 | 13.7 | 13.6 | 13.7 |