# OpenReview forum: "Unsupervised Speech Separation Using Mixtures of Mixtures"
_ICML.cc/2020/Workshop/SAS — SAS 2020_

### Official Review · AnonReviewer1 · 2020-06-27
**An interesting step towards unsupervised speech separation**

**Rating:** 9
**Confidence:** 3

**Review:**

The paper proposes a solution for unsupervised speech separation. The work, instead of using the classical approach based on simulating mixture starting from clean signals, it creates mixtures of mixtures. In practice, two mixtures containing up to N sources are combined and the neural network is trained to predict 2N outputs. To make it possible, a modification to the standard permutation invariant training has been proposed. The results clearly show the effectiveness of the proposed approach, especially when combined with some examples trained with the standard supervised approach.

To the best of my knowledge, the idea is novel. The paper is also very well written and clearly states the given contribution compared to previous works.  The topic addressed is definitely in line with the topic of the workshop.

Additional comments:
- One argument used by the authors is that the creating mixture using the standard simulated approach is nor that realistic. This is true, but one might argue that even combing two independent mixtures (potentially corrupted by different noises, captured with different microphones,..) is not realistic too.
- As pointed out in the paper, one issue is to overestimate and underestimate the sources. The part where the authors discuss why underestimation is discouraged is not totally clear to me and I would encourage the authors to revise a bit that part.

---

### Official Review · AnonReviewer3 · 2020-06-29
**Interesting work on a not yet fully explored topic.**

**Rating:** 8
**Confidence:** 5

**Review:**

This paper discusses an unsupervised/semi-supervised approach for neural network-based speech separation. It is based on the simple idea of mixing two observed mixtures of unknown sources and training the system so that the remixed outputs obtained from separating the mixed mixture re-create the two observed mixture signals. The authors showed that networks trained with this approach can learn to separate without any reference clean speech signal. The approach can also be used in a semi-supervised scheme. The paper is relatively easy to follow except for some parts of the descriptions of the experiments. The authors should however add the missing important reference mentioned below.
Pros:
The paper presents an interesting idea for an important, not well-covered topic.
Cons:
The main issue is the lack of reference to a recent work that tackles the same problem and has is much related to this study.
Besides, the description of the experiments is sometimes unclear.

Below are some additional comments.
1.	There is a reference missing to a much related recent work that proposed a semi-supervised approach for speech separation. It is hard to believe that none of the authors noticed that work, and I am very curious what the reasons for not citing it are.
Although there are clear differences, the authors should at least cite this work and mention clearly the differences as both works use a remixing strategy and consistency loss, although the work below does not use mixtures of mixtures and would not work in a fully unsupervised setting.
Ideally, it would be better to compare both methods in the experiments (at least for the semi-supervised conditions), but that may be asking too much for a conference paper.
M. W. Y. Lam, J. Wang, D. Su and D. Yu, "Mixup-breakdown: A Consistency Training Method for Improving Generalization of Speech Separation Models," ICASSP 2020 - 2020 IEEE International Conference on Acoustics, Speech, and Signal Processing (ICASSP), Barcelona, Spain, 2020, pp. 6374-6378, doi: 10.1109/ICASSP40776.2020.9054719.
2.	Figure 2 is a little complex to understand because it includes in a compact manner many different conditions. It thus requires some time to understand what corresponds to fully unsupervised conditions etc. It thus requires some time to understand what corresponds to fully unsupervised conditions etc.
I assume that the left bars correspond to the fully unsupervised case (100% matched unsupervised data). In this case, there should not be any difference between the results of “Matched supervised, 1 or 2 sources” and “Mismatched supervised, 1 or 2 sources”, because no supervised data is actually used, right? However, the results show a small difference. Is it due to some randomness in the models, or did I miss something? Could you make this point clearer?
3.	It is surprising that when training with “1 or 2 sources” conditions, supervised data do not provide any gains, but rather than to degrade performance. Could you comment more on that? Does that mean that we shouldn’t use any supervised data?
4.	Could you clarify what mismatched supervised data you used for each model?
5.	There are several practical issues that would deserve more discussions such as how to handle noisy mixtures and what would happen if the same speaker appears in both mixtures in some cases? I hope that these will be addressed in future works.

---

### Official Review · AnonReviewer2 · 2020-06-30
**A well-written paper with a novel unsupervised approach for singe-channel speech separation**

**Confidence:** 4
**Rating:** 9

**Review:**

The paper introduces a novel approach to tackle single-channel speech separation in an unsupervised fashion. The high-level idea is to leverage a mixture of mixtures with permutation-invariant training.

Pros:
 - The idea is novel and neat
 - The paper is well-written
 - The conducted experiments demonstrate the effectiveness of the approach, which is on par with supervised training. The semi-
   supervised results are interesting.

Cons:
 - Lack of human evaluation
 - It's not clear if the approach is scalable or not when there are more than 2 speakers.

---

### Decision · Program_Chairs · 2020-07-01

**Decision:**

Accept

**Comment:**

Dear author(s),

Thank you very much for your submission at the ICML2020@SaS workshop (https://icml-sas.gitlab.io/). Based on the scores assigned by the reviewers, we are happy to notify you that your paper was accepted for the workshop.

Please, address the comments of the reviewers and submit the camera-ready version by July 8. We ask the authors to record a 15min video for your talk. At the workshop, we will have the pre-recorded video as well as a live QA session. It is important to keep this time limit, otherwise, your talk will be automatically cut. The deadline for uploading the video is July 8. The detailed instructions for uploading will follow.

Feel free to contact us for any questions!

Best,

The ICML20@SaS organizers:
Mirco Ravanelli
Titouan Parcollet
Dmitriy Serdyuk
Devon Hjelm
Bhuvana Ramabhadran